# Terrain Perception Using Wearable Parrot-Inspired Companion Robot, KiliRo

**DOI:** 10.3390/biomimetics7020081

**Published:** 2022-06-14

**Authors:** Jaishankar Bharatharaj, Loulin Huang, Ahmed M. Al-Jumaily, Senthil Kumar Sasthan Kutty, Chris Krägeloh

**Affiliations:** 1Institute of Biomedical Technologies, Auckland University of Technology, Auckland 1010, New Zealand; ahmed.aljumaily@aut.ac.nz; 2PAIR LAB, Bharath Institute of Higher Education and Research, Chennai 600073, India; director.robotics@bharathuniv.ac.in; 3PAIR LAB, Auckland University of Technology, Auckland 1010, New Zealand; loulin.huang@aut.ac.nz (L.H.); chris.krageloh@aut.ac.nz (C.K.)

**Keywords:** wearable robot, parrot inspired robot, terrain perception, KiliRo

## Abstract

Research indicates that deaths due to fall incidents are the second leading cause of unintentional injury deaths in the world. Death by fall due to a person texting or talking on mobile phones while walking, impaired vision, unexpected terrain changes, low balance, weakness, and chronic conditions has increased drastically over the past few decades. Particularly, unexpected terrain changes would many times lead to severe injuries and sometimes death even in healthy individuals. To tackle this problem, a warning system to alert the person of the imminent danger of a fall can be developed. This paper describes a solution for such a warning system used in our bio-inspired wearable pet robot, KiliRo. It is a terrain perception system used to classify the terrain based on visual features obtained from processing the images captured by a camera and notify the wearer of terrain changes while walking. The parrot-inspired KiliRo robot can twist its head and the camera up to 180 degrees to obtain visual feedback for classification. Feature extraction is followed by K-nearest neighbor for terrain classification. Experiments were conducted to establish the efficacy and validity of the proposed approach in classifying terrain changes. The results indicate an accuracy of over 95% across five terrain types, namely pedestrian pathway, road, grass, interior, and staircase.

## 1. Introduction

The World Health Organisation estimates that globally about 684,000 individuals die every year due to falls [1]. Most fall incidents are due to people texting or talking on mobile phones while walking, impaired vision, unexpected terrain changes, low balance, weakness, and chronic conditions. Mobile phone usage has increased drastically over the past few decades and is expected to reach over 7.6 billion users by 2027 [1]. With 62.9% of the world’s population already owning a mobile phone, it is reported that young adults spend an average of five hours a day using their mobile phones [2,3]. With reported global revenue of about USD 450 billion through smartphone sales in 2021 [4], it is predicted that this industry will dominate the global market in many areas, including marketing, sales, media, and communication. Despite numerous benefits, the exponential increase in the usage of mobile phones has also reported several concerns, such as health issues and accidents. Usage of mobile phones while walking is one of the main concerns in recent years, i.e., 5754 emergency cases were related to mobile phone usage while walking between 2000 and 2011 in the US [5]. Research shows that about 500,000 drivers may be using mobile phones while driving at any given time in the United States [6] and 43% of pedestrians meet with accidents due to mobile phone usage [7], which most of the time leads to bumping into walls, collision between pedestrians, falling, and overlooking traffic signals. Several cases reported fall incidences due to mobile phone usage while walking which at times leads to severe injuries and sometimes death [8]. Impaired vision also causes serious fall incidences leading to serious physical damage. The literature indicates that poor visual acuity, self-reported poor vision, visual field loss, poor stereoscopic vision, and impaired contrast sensitivity are significantly associated with fall incidences [9]. Particularly, older adults are at high risk of falls due to vision impairment. The CDC states that there are over twelve million Americans with vision impairment and the number is over 2.2 billion globally [10]. This huge population is at high risk of falls and demands an innovative approach to a safe environment. Similarly, low balance, weakness, and chronic conditions cause falls and serious damage to individuals. While the majority of fall incidences are due to health conditions, unexpected terrain changes can cause falls even among any individual.

In this article, we aim to predict and warn individuals about sudden terrain changes, one of the main causes of fall incidences as pedestrians are unaware of these transitions. A wearable robot with the ability to warn pedestrians about terrain changes can be very useful in avoiding such fall incidences.

The applications of wearable robots are manifold, including healthcare, education, companionship, entertainment, security, and lifelogging. For example, Rahman et al. developed an exoskeleton robot, a class of wearable robots for the rehabilitation of elbow and shoulder joint movements [11]. The designed robot, ExoRob, is worn on the lateral side of the upper arm to provide naturalistic movements at the level of elbow and shoulder joint rotation. Similarly, there are several studies that discuss the design, development, and deployment of wearable robots in the healthcare industry [12,13,14]. Similarly, the deployment of such robots has been explored extensively for personal needs. As an illustration, Kostov, Ozawa, and Matsuura developed a wearable accessory robot for context-aware apprise of personal information and showed that the development of a wearable robot is an important step toward a new era of wearable computing [15]. In another study, a wearable robotic device to enhance social interactivity and provide an emotionally immersive experience for real-time messaging was proposed by Dzmitry and Alena. This device helped reinforce the personal feelings of the wearer and reproduced the emotions felt by the partner during online communication [16]. Assisting the visually impaired is an important area of application for wearable robots. Several studies have been conducted to develop wearable devices for assisting them in transportation, communication, and education [17,18,19]. Researchers at the University of Southern California presented a head-mounted, stereovision-based navigational assistance device for the visually impaired. The developed head-mounted robot enables the wearer to stand and scan the scene for integrating wide-field information [20].

Even though the application of wearable robots has been studied in detail for several years, including the application of pedestrian safety bio-inspired robots has never been explored in this context. A wearable robot capable of performing terrain perception can be very useful in alerting the pedestrian when sudden terrain changes are noted. Such robots can largely help reduce fall incidences and minimize accidents.

Terrain perception and classification have been studied extensively in several fields, including security, rescue, and service. Particularly, terrain traversability analysis is widely used in unmanned ground vehicles for safe navigation and avoiding collisions [21]. Michael and Sabastian addressed the problem of terrain modeling in robot navigation by proposing an approach by acquiring a set of terrain models at different resolutions [22]. This approach produced significantly better results in a practical robot system, capable of acquiring detailed 3D maps on a large scale. In another study, Dominik and Piotr presented an integrated system for legged robot navigation in previously unseen and uneven terrain using onboard terrain perception [23].

Service robotics is another important area where terrain perception is applied extensively. As an illustration, robotic wheelchairs available in the market use various terrain perception methods to detect and avoid unsafe situations for the rider. Volodymyr et al. developed a system for guiding visually impaired wheelchair users along a clear path using computer vision to avoid obstacles and warn the rider about terrain changes [24]. With such extensive application of terrain perception and classification methods in robotics, we are unaware of any study that uses this method in wearable robots. Applying terrain perception and classification in a wearable robot can contribute extensively to reducing fall incidences and avoiding accidents related to the usage of mobile phones while walking.

Presently, there are several initiatives taken around the world to ensure pedestrian safety. A comprehensive pedestrian safety plan was announced in New York in 2016 [25]. This multi-agency initiative has provided USD 110 million to improve safety for pedestrians through infrastructure improvements, public education efforts, and enforcement across the state. Displaying pedestrian safety signage, ads, and conducting safety lessons and campaigns are a few strategies to educate pedestrians on their safety. However, about 6000 pedestrian deaths due to distraction have been recorded in the U.S. in 2017 [26]. This alarming report demands innovative approaches to warn pedestrians of dangers while walking. Hence, the proposed wearable bio-inspired robot, KiliRo, can be useful to warn the wearer of terrain changes, thus avoiding fall incidents and related accidents. Though the KiliRo robot has been used in therapeutic settings and reported success in improving the learning and social interaction of children [27,28,29], this is the first time the parrot-inspired wearable robot has been designed for helping pedestrians. Biologically inspired robots have shown to be beneficial in several applications, including therapeutic and service needs [30]. Particularly, human–parrot relationships have existed for several centuries. There are references to parrots being used as a companion and messenger in Tamil literature from several centuries ago [31,32]. The goddess Meenakshi statue at an ancient Hindu temple in Madurai, India, built from 1190 CE to 1205 CE, is another example of the human relationship with parrots. In this statue, the goddess’s raised hand holds a lotus, on which sits a green parrot [33]. Parrots have helped humans in communication, companionship, and therapeutic needs enormously. Hence, a robot designed through inspiration from a parrot can help increase human–robot interactions and robot acceptance in humans.

In this paper, we present a terrain perception and classification system for our novel parrot-inspired wearable robot, KiliRo, to classify five types of terrains based on visual features and warn the wearers of sudden terrain changes. Our proposed approach uses a speeded-up robust feature (SURF) description along with color information. The feature extraction will be followed using K-nearest neighbor for terrain classification.

To facilitate the reading comprehension of the article, the remainder of the paper has been organised as follows. First, in Section 2, we describe the design and develop-ment of the wearable parrot-inspired robot, KiliRo and illustrate the methods used for feature extraction and classification of images captured using the KiliRo robot for ter-rain classification. Section 3 presents the experimental setup and the results obtained. In Section 4, we discuss the potential for deploying the wearable parrot-inspired robot to warn users on sudden terrain changes and acknowledge the limitations and need for further research in this context. Finally, in Section 5, we conclude with the research findings and present our future works.


## 2. Materials and Methods

The main scope of this research is to develop a terrain perception and classification system for the wearable robot, KiliRo, that can warn the wearer of sudden terrain changes when transitions are identified. In terms of morphology, the KiliRo robot can be defined as a two-legged wearable robot, having a physical appearance that resembles a parrot.

We considered a set of design constraints in deciding the dimensions of the robot during the concept generation process:height < 250 mm;weight < 250 g;head rotation range: 180°;operate between 10° and 45° Celsius.

After a series of brainstorming sessions on concept generation and selection sessions, we developed a new version of the KiliRo robot aimed at achieving 180° of a rotating head design. The curvature of the robot’s leg design was optimized to create a wearable robot design. The dimensions of the newly developed KiliRo and the selection of commercial devices, such as servo motors and electronic boards, were opted to fit the robot design constraint on size and weight. The robot has three parts: head, body, and wings. The neck part connects the head and body. A static tail is attached at the top of the head for aesthetic appeal. The robot’s three parts are designed to be hollow to minimize the weight and optimize the three-dimensional printed materials. The specifications of the mechanical properties of the wearable parrot robot are listed in Table 1.

The robot’s head is mounted with two servo motors (SG90, manufactured by Tower Pro) to provide Pitch and Yaw motions. The robot can turn its head 90° left and right from the center and move up and down. In other words, the produced design can rotate 180° along the x-axis and move 45° along the y-axis. This locomotion allows the robot to look for terrain changes while the wearer is walking. The exploded view of the KiliRo robot is presented in Figure 1 where the detailed design of the robot parts and joints are demonstrated. The head rotation positions are illustrated in Figure 2.

The initial head position of the robot is set at 0° and, when initiated, the head rotates along the x-axis and y-axis following a predefined cycle to detect terrain changes. The robot uses the camera mounted on its head for terrain perception and classifications. The hardware used in the robot is listed in Table 2.

### 2.1. Feature Extraction and Classification

#### 2.1.1. SURF

Prior to training the classifier, obtaining information from the training data that are useful in classification is recommended. These features can provide a distinction between different classes. In this paper, we use speeded-up robust features (SURF) for feature extraction. SURF is a patented local feature detector and descriptor used for object recognition, image registration, classification, or 3D reconstruction [34].

As the name suggests, this feature extraction is much faster than many other methods such as scale-invariant feature transform (SIFT), features from accelerated segment test (FAST), and principal components analysis-scale-invariant feature transform (PCA-SIFT) [35]. The first step is to find the area of interest which is performed using the blob detector based on the determinant of Hessian, a square matrix of second-order partial derivatives of a scalar-valued function, or scalar field. Due to its computation time and accuracy, the Hessian matrix is used in SURF and the approach is used for selecting the location and the scale. The Hessian matrix describes the local curvature of a function of variables. The interest point is then sampled into a 4 × 4 grid thus separating the region into 16 square grids. Now, the HAAR wavelets (a sequence of rescaled square-shaped functions which together form a wavelet family or basis) are extracted from each of the grids at 5 × 5 regularly spaced sample points. The feature descriptor is obtained by summing up all the HAAR wavelets obtained from all 16 grids. In this experiment, the feature descriptors obtained through the above method are used for classification purposes.

#### 2.1.2. KNN

Now that the features have been extracted, the next step is to classify the images based on the descriptors and keywords obtained from the SURF features. The classification method used in this paper is K-nearest neighbor (KNN). KNN is a supervised learning algorithm, and the classification is based on finding out the K number of neighbors and their corresponding class. One reason behind using this classification strategy is that this algorithm supports multi-class classification and provides a much better accuracy than many other algorithms. Moreover, it allows the user to modify the classification strategy based on a different scenario by changing the ‘K’ value. Again, the KNN presented in the OpenCV python package is used for this purpose.

The basic idea behind the K-NN is to find out the distance between the new test point and the training data. The closer the test point is to a training data, the more the possibility that the test point belongs to the respective class. The user must mention the value of K to begin the algorithm. The algorithm then calculates the K number of training data that are closest to the new test data. Then, the new test data belong to the class with the greatest number of its training data within the ‘K’ nearest neighbor. For example: in the case of k = 3, if class 1 has two training data within the 3 nearest neighbors of the test data but class 2 has one training data within the 3 nearest neighbors of the test data, then the new test data belong to class 1.

There are two decisions to be made during the image classification phase: One is the value of K and the second one is the distance formula to be used for the calculation. The value of K for this experiment is chosen to be one. Increasing the number of K increases the accuracy but it could also, in turn, increase the time consumption and complexity of the program. We can obtain a good result with the K value of one. In terms of distance calculation, the Euclidean distance definition is adopted.
(1)$$D\left(x,y\right)=\sqrt{\overset{N}{\underset{i=1}{\sum }}{\left(X_{i}-Y_{i}\right)}^{2}}$$
where
*N* is the dimension of the feature.*D* (*x*, *y*) gives the Euclidean distance between points *X* and *Y*.*X_i_* refers the *i^th^* feature of *X*.*Y_i_* refers to the *i^th^* feature of *Y*.

The diagram illustrating the working of KNN is shown in the following Figure 3.

The red question mark is the new test data. Now consider the case of k = 3, the new data belong to the blue class, whereas, in the case of k = 5, the new data belong to a green class. The flowchart for implementing the terrain classification in the KiliRo robot is shown in Figure 4.

#### 2.1.3. Experimental Setup

In our experiments, we considered five types of terrains, namely grass, pathway, road, staircase, and interior. All terrains are on the campus of the Singapore University of Technology and Design (SUTD), Singapore. As the efficiency of terrain perception and classification depends on the effectively built database, we captured 226 images in different lighting and background settings. After the database for terrain perception was established, the terrain classification was performed using the SURF and KNN methods detailed in previous sections. The key points obtained from each of the different class are shown below in Figure 5.

A picture describing how the robot would be attached to the user is shown in Figure 6. In this picture, a tripod is used to attach the robot to the user’s shoulder and can move its head to capture images of terrains.

## 3. Results

A research assistant acting as a user was instructed to walk for a few minutes in a predefined path of five types of terrains namely, grass, pathway, road, staircase, and interior space to capture images at different angles. The session was continued until the required images for terrain classification of each space were captured. Two hundred and twenty-six pictures were taken by the parrot robot at a rate of 60 frames per second while the user wore the robot and was exposed to different terrains. These images were then analyzed using the developed terrain perception and classification system. The results are illustrated in Table 3.

The above results report an accuracy value of 100% for four terrains and an accuracy of above 95% for one terrain, which puts the overall accuracy of the classifier at 99.02%.

## 4. Discussion

While the literature points to several fall incidences resulting in severe injuries and sometimes death, an approach to warn pedestrians of sudden terrain changes can help reduce such accidents. Wearable robots capable of detecting terrains and sudden changes in them can help reduce fall incidences by warning the wearers.

In this study, we developed a wearable parrot robot and a terrain perception and classification model to classify five types of terrains. These two are separate tasks: (1) 226 pictures of five types of terrains were taken using the wearable parrot robot; (2) these images were tested using the terrain perception and classification system and analyzed the efficiency of the developed system. This is a pioneering study to develop a wearable parrot-inspired robot and use pictures taken by the parrot robot for terrain perception and classification. While the terrain perception approach has been extensively studied in mobile robots [36,37] and bio-inspired robots [38], we aim to develop a terrain perception and classification system for the novel wearable parrot-inspired robot to warn the wearers of sudden terrain changes. In our future studies, we will implement the terrain perception and classification system in real time to warn the user of sudden terrain changes using voice commands. This robot can be used in different applications including assisting the visually impaired, elderly, pedestrians, and children. Our future research on the deployment of a wearable robot includes assisting school children in classrooms to record and play back lectures and entertain and investigate psychological changes of the wearer.

There are identified limitations in this study. First, we have used five pre-decided types of terrains and novel images that were not used to test if the proposed terrain perception and classification system was efficient. Second, the pictures were taken when the wearer of the robot was stable and not moving, as pictures taken during movement could affect the quality of the image and thus the feature extraction. Third, real-time classification should be performed to identify the time delay in classifying the terrains and intimating the wearer. Fourth, the time taken to classify the terrain was not investigated. Last, the present morphology of the robot uses a tripod to attach the robot to the wearer and the user may not feel comfortable with this external attachment. These limitations are noted, and our future studies will minimize them.

## 5. Conclusions

In this paper, we presented a new approach for the vision-based terrain perception and classification system for a wearable pet robot, KiliRo, which is based on speeded-up robust features and K-nearest neighbor methods. The image features from the pictures taken from the KiliRo robot were extracted using the SURF feature extraction method. As mentioned in the experimental setup, the key points and descriptors for each terrain are stored in the program. When a test image is uploaded, the algorithm uses the KNN classification method to compare the test image to the extracted features from each of the terrains. The terrain feature which has the closest neighbor to the features extracted from the test image is the output and that class is chosen to be an appropriate classification. The proposed system for the safety of pedestrians reported an accuracy of 99.02 percent. Future work would focus on the integration of additional sensors to further improve the terrain perception and classification for more terrains beyond what is currently possible. Another work would be to improve the aesthetic appeal of the KiliRo robot and conduct the study with real-time users.

## Figures and Tables

**Figure 1 biomimetics-07-00081-f001:**
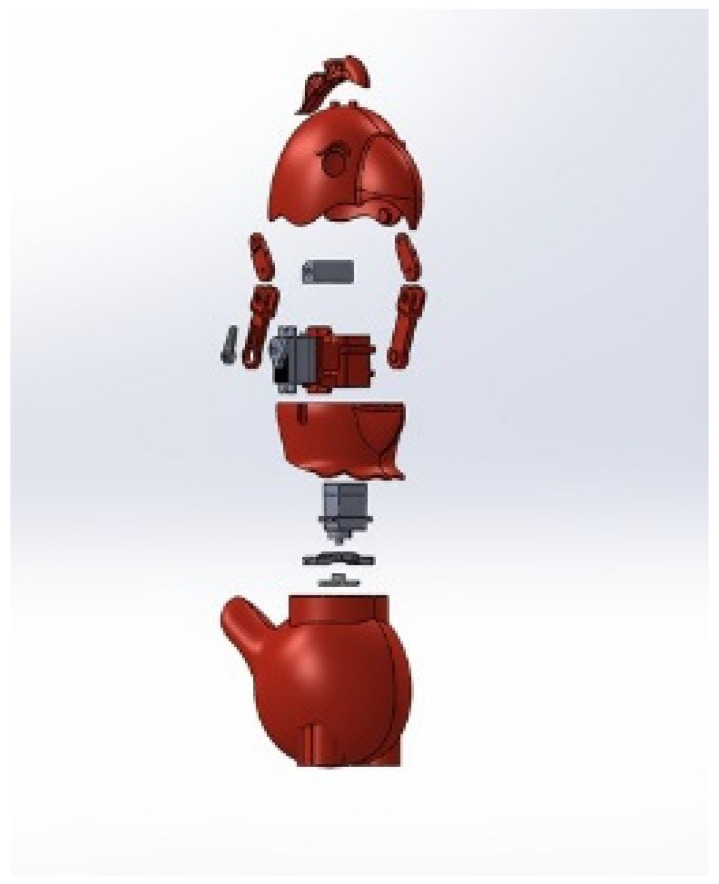
Wearable KiliRo robot—motor positions.

**Figure 2 biomimetics-07-00081-f002:**
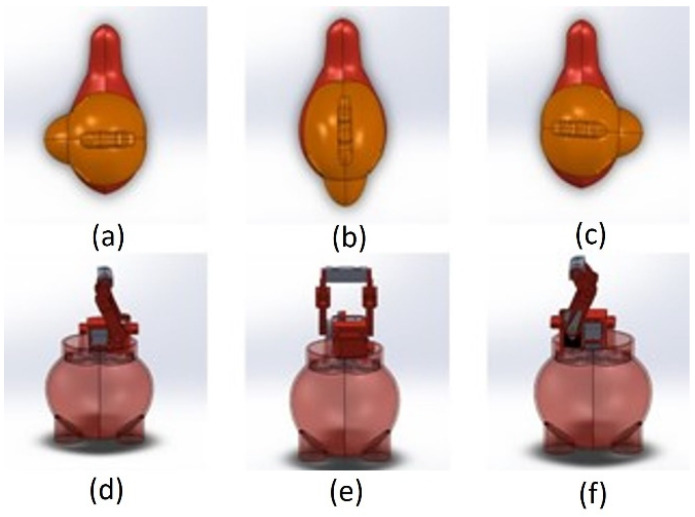
KiliRo robot head orientations. (**a**) right-side view; (**b**) straight view; (**c**) left-side view; (**d**) internal view of right-side (**e**) internal view of straight (**f**) internal view of left-side.

**Figure 3 biomimetics-07-00081-f003:**
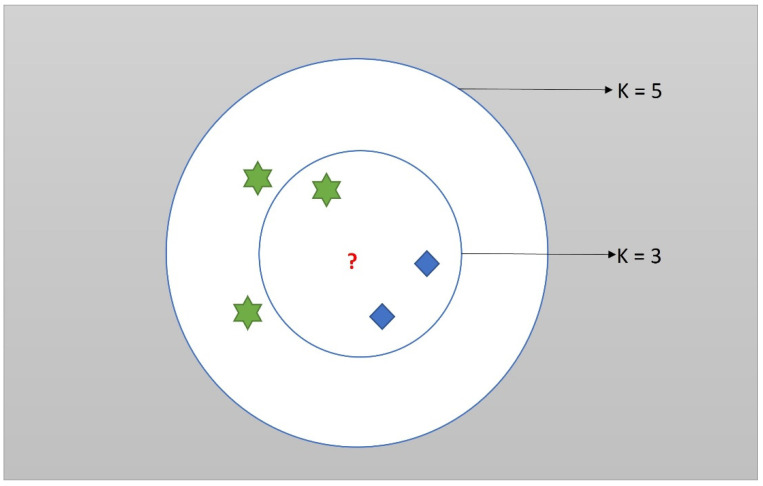
The red question mark is the new test data. Now consider the case of k = 3, the new data belong to the blue class whereas, in case of k = 5, the new data belong to a green class.

**Figure 4 biomimetics-07-00081-f004:**
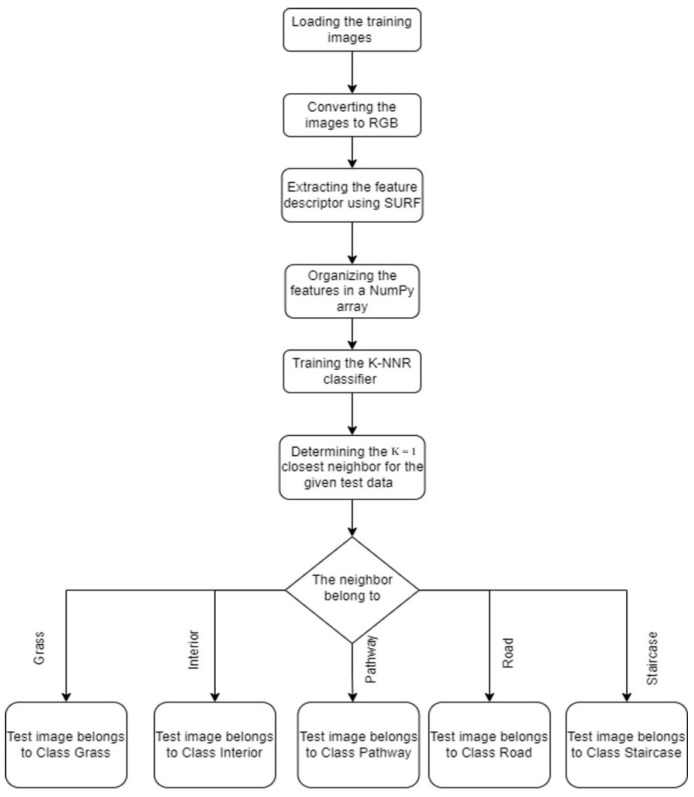
Flowchart of the algorithm used for terrain classification.

**Figure 5 biomimetics-07-00081-f005:**
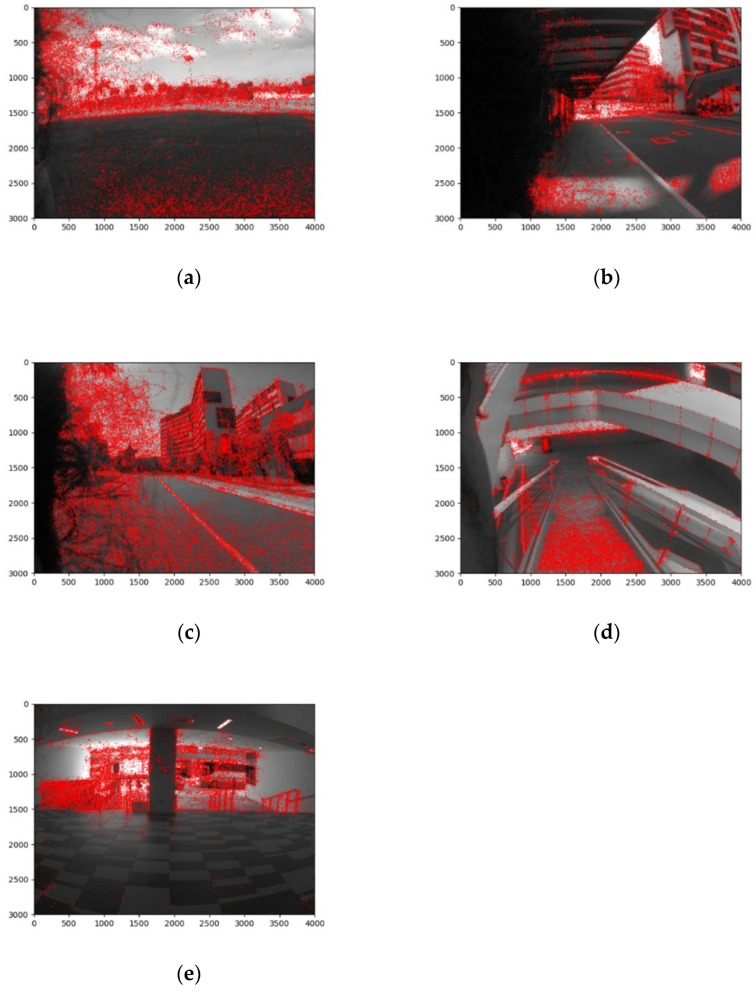
Key points for the five classes obtained by the KiliRo robot. (**a**) Grass; (**b**) pathway; (**c**) road; (**d**) staircase; (**e**) interior.

**Figure 6 biomimetics-07-00081-f006:**
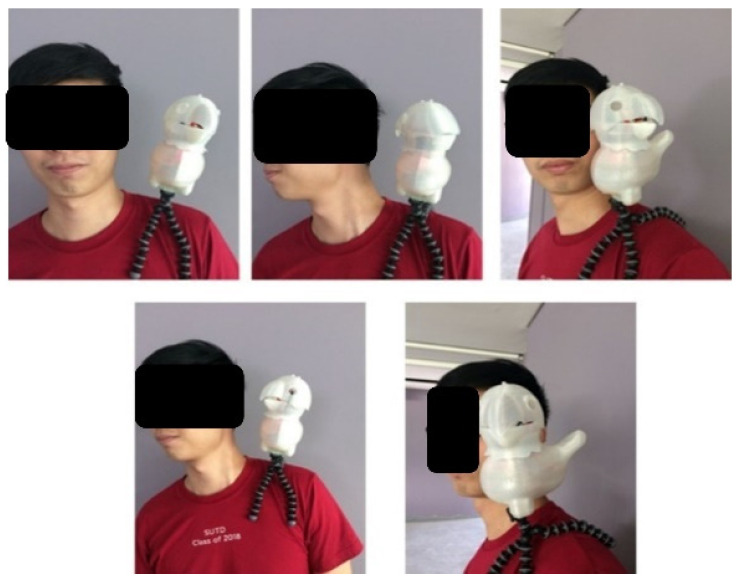
Various positions and orientations of the KiliRo robot.

**Table 1 biomimetics-07-00081-t001:** Specifications of the mechanical properties of wearable parrot robot.

Robot Body Material	PLA (Poly Lactic Acid)
Dimensions W × H	160 mm × 80 mm
Weight	140 g
Head rotation	180°
Head tilt	45°

**Table 2 biomimetics-07-00081-t002:** Robot hardware specifications.

Hardware	Specification
Controller	Raspberry pi
Servo motor	TowerPro SG90
Servo controller	Pololu-Micro Maestro 18-channel USB servo controller
Camera	Ai—ball camera
Battery	Li-Po 1200 mAh 7.4v
Power regulator	Dimension Engineering De-SW033

**Table 3 biomimetics-07-00081-t003:** Table indicating the terrain classification results for the KiliRo robot.

Terrain	Grass	Interior	Pathway	Road	Staircase	Accuracy
Grass	45	0	0	0	0	100
Interior	0	35	0	0	0	100
Pathway	2	0	39	0	0	95.12
Road	0	0	0	45	0	100
Staircase	0	0	0	0	62	100
Recall (%)	95.74	100	100	100	100

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
