# Peer review of "Terrain Perception Using Wearable Parrot-Inspired Companion Robot, KiliRo"

_biomimetics, 2022, doi:10.3390/biomimetics7020081_

Round 1
Reviewer 1 Report
This paper proposes a warning system to alert people of the imminent danger of fall.
The analysis is reasonable and the design is perfect. Some comments are given below:
1. Figure 1 and Figure 2 about the KiliRo robot is not clear, and it is better to mark the name of each part in the picture.
2. The working principle of the KiliRo robot based on Parrot bionic design is not clear.
3. How long does the training model identify the terrain?
Author Response
The analysis is reasonable and the design is perfect.
Thank you very much for the positive comments.
- Figure 1 and Figure 2 about the KiliRo robot is not clear, and it is better to mark the name of each part in the picture.
We updated the manuscript.
- The working principle of the KiliRo robot based on Parrot bionic design is not clear.
Thank you for the comment. Our approach is not mimic the working model of parrots but to inspire its morphology. We conducted an extensive user requirement analysis before we conclude the features and specifications of our robot which can be found in “Bharatharaj, J. (2018). Investigation into the development of a parrot-inspired therapeutic robot to improve learning and social interaction of children with autism spectrum disorder (Doctoral dissertation, Auckland University of Technology)”.
- How long does the training model identify the terrain?
We conducted test-retest method and the results are illustrated in our results section. But, we have not timed the terrain classification process. We included this detail in the manuscript as a limitation.
Reviewer 2 Report
1.The author should be check the paper carefully and put the relevant references analysis into the "Related work" subsection.
2. The line 85 and 86 in the paper should be revised. There are several works have been done on the pedestrian safety using similar robots.
3.The author should add more detailed related work analysis.
4.The line 197 in the paper should be revised. The abbreviation of the K-Nearest Neighbor is not KNNR.
5.The line 223 in the paper should be revised. The "Figure 4" should be "Figure 3".
6.The line 230 in the paper should be revised. The "Figure 3" should be "Figure 4".
7. The line 242 in the paper should be revised. The "Figure 2" should be "Figure 5".
8.The line 248 in the paper should be revised. The "Figure 1" should be "Figure 6".
9.The program running frame rate should be added into the "Results" subsection.
Author Response
1.The author should be check the paper carefully and put the relevant references analysis into the "Related work" subsection.
Thank you. We included references based on other reviewers’ comments as well.
- The line 85 and 86 in the paper should be revised. There are several works have been done on the pedestrian safety using similar robots.
We changed the phrase. Thank you for the comments.
3.The author should add more detailed related work analysis.
References included.
4.The line 197 in the paper should be revised. The abbreviation of the K-Nearest Neighbor is not KNNR.
Changed the content. Thank you for the notification.
5.The line 223 in the paper should be revised. The "Figure 4" should be "Figure 3".
Apologies for the typo. Information updated.
6.The line 230 in the paper should be revised. The "Figure 3" should be "Figure 4".
Apologies for the typo. Information updated.
- The line 242 in the paper should be revised. The "Figure 2" should be "Figure 5".
Apologies for the typo. Information updated.
8.The line 248 in the paper should be revised. The "Figure 1" should be "Figure 6".
Apologies for the typo. Information updated.
9.The program running frame rate should be added into the "Results" subsection.
Information included. Thank you!
Reviewer 3 Report
This is an interesting paper. For bio-inspired design, the authors are expected to discuss the biological advantage of this design. Some bio-inspired algorithm is also preferable, e.g., beetle antennae search for various optimizations. Please discuss this possibility in the introduction part on using bio-inspired algorithms for this new robot.
Author Response
This is an interesting paper. For bio-inspired design, the authors are expected to discuss the biological advantage of this design. Some bio-inspired algorithm is also preferable, e.g., beetle antennae search for various optimizations. Please discuss this possibility in the introduction part on using bio-inspired algorithms for this new robot.
Thank you for the positive comments. We have included the discussion as recommended.
Reviewer 4 Report
Review of: biomimetics-1709614
Terrain perception in mobile robots is not new [Wolfe, 1989], [Bellutta et al., 2000] and more recently [Zhu Hongwu et al., 2021]. The use of nearest neighbour algorithms is also not unique [Masataka et al., 2016]. Terrain perception in biomimetic robots are also described in the literature [Sinha et al., 2014].
The strength of the paper lies mainly in its application. However, the relevance to biomimetics is scanty. Such a device in the form of a parrot may appeal to children (which, of course, is the main application) but it could also simply be in the form of a wearable camera.
More specifically, there appears to be a lack of analysis. For example, the authors mention the use of the Hessian matrix but do not explain how it is employed. The Hessian matrix can be used to determine local maxima and minima, saddle points etc. Which of these features are used?
Although the English is adequate, there are many small grammatical errors which need to be removed. For example, the last phrase in the conclusions “...conduct t the study with real-time users”.
Additional references
P. Bellutta, R. Manduchi, L. Matthies, K. Owens and A. Rankin, "Terrain perception for DEMO III," Proceedings of the IEEE Intelligent Vehicles Symposium 2000 (Cat. No.00TH8511), 2000, pp. 326-331, doi: 10.1109/IVS.2000.898363
Zhu Hongwu, Wang Dong, Boyd Nathan, Zhou Ziyi, Ruan Lecheng, Zhang Aidong, Ding Ning, Zhao Ye, Luo Jianwen. “Terrain-Perception-Free Quadrupedal Spinning Locomotion on Versatile Terrains: Modeling, Analysis, and Experimental Validation” Frontiers in Robotics and AI, Vol 8, 2021, doi: 10.3389/frobt.2021.724138
William J. Wolfe (Ed.) “Mobile Robots III”, Vol 1007. Chapter “Perception For Rugged Terrain” In So Kweon and Martial Hebert and Takeo Kanade, pp 103-119, International Society for Optics and Photonics (SPIE), 1989. doi:10.1117/12.949089
Masataka, Fuchida and Mohan, Rajesh Elara and Tan, Ning and Nakamura, Akio and Pathmakumar, Thejus. “Terrain Perception in a Shape Shifting Rolling-Crawling Robot”. Robotics, Vol 5, 2016. https://www.mdpi.com/2218-6581/5/4/19
Sinha, A., Tan, N. & Mohan, R.E. Terrain perception for a reconfigurable biomimetic robot using monocular vision. Robot. Biomim. 1, 23 (2014). https://doi.org/10.1186/s40638-014-0023-2
Author Response
Terrain perception in mobile robots is not new [Wolfe, 1989], [Bellutta et al., 2000] and more recently [Zhu Hongwu et al., 2021]. The use of nearest neighbour algorithms is also not unique [Masataka et al., 2016]. Terrain perception in biomimetic robots are also described in the literature [Sinha et al., 2014].
Thank you for the comments. While we do not claim novelty in terrain perception or the use of KNN, we realise that Line131 does sounds like. We changed those lines and made our claim clear.
The strength of the paper lies mainly in its application. However, the relevance to biomimetics is scanty. Such a device in the form of a parrot may appeal to children (which, of course, is the main application) but it could also simply be in the form of a wearable camera.
Thank you for the comment. Our approach is to improve human-robot interaction through companion robots. Our KiliRo robot has previously showed to improve learning and social interaction of children with autism. We also proved that the parrot-inspired robot can significantly reduce the stress levels of these children. The current approach is to design a wearable companion robot which can provide several benefits including the proposed warning system. While a camera can achieve the task, it may not provide other benefits we are targeting.
More specifically, there appears to be a lack of analysis. For example, the authors mention the use of the Hessian matrix but do not explain how it is employed. The Hessian matrix can be used to determine local maxima and minima, saddle points etc. Which of these features are used?
Thank you. We have included the details.
Although the English is adequate, there are many small grammatical errors which need to be removed. For example, the last phrase in the conclusions “...conduct t the study with real-time users”.
We apologies for the typo. The whole manuscript has been reviewed for errors. Thank you.
Round 2
Reviewer 4 Report
although the improvements are minimal, the paper is possibly acceptable now.